# SARS-CoV-2 Infectivity and Severity of COVID-19 According to SARS-CoV-2 Variants: Current Evidence

**DOI:** 10.3390/jcm10122635

**Published:** 2021-06-15

**Authors:** Thi Loi Dao, Van Thuan Hoang, Philippe Colson, Jean Christophe Lagier, Matthieu Million, Didier Raoult, Anthony Levasseur, Philippe Gautret

**Affiliations:** 1IRD, AP-HM, SSA, VITROME, Aix Marseille University, 13005 Marseille, France; thiloi.dao@gmail.com (T.L.D.); thuanytb36c@gmail.com (V.T.H.); 2IHU—Méditerranée Infection, Aix Marseille University, 13005 Marseille, France; philippe.colson@univ-amu.fr (P.C.); jean-christophe.lagier@univ-amu.fr (J.C.L.); matthieumillion@gmail.com (M.M.); didier.raoult@gmail.com (D.R.); anthony.levasseur@univ-amu.fr (A.L.); 3Thai Binh University of Medicine and Pharmacy, Thai Binh 410000, Vietnam; 4IRD, AP-HM, MEPHI, Aix Marseille University, 13005 Marseille, France

**Keywords:** SARS-CoV-2, COVID-19, mutants, variants, infectivity, severity

## Abstract

Background: We conducted this review to summarize the relation between viral mutation and infectivity of SARS-CoV-2 and also the severity of COVID-19 in vivo and in vitro. Method: Articles were identified through a literature search until 31 May 2021, in PubMed, Web of Science and Google Scholar. Results: Sixty-three studies were included. To date, most studies showed that the viral mutations, especially the D614G variant, correlate with a higher infectivity than the wild-type virus. However, the evidence of the association between viral mutation and severity of the disease is scant. A SARS-CoV-2 variant with a 382-nucleotide deletion was associated with less severe infection in patients. The 11,083G > U mutation was significantly associated with asymptomatic patients. By contrast, ORF1ab 4715L and S protein 614G variants were significantly more frequent in patients from countries where high fatality rates were also reported. The current evidence showed that variants of concern have led to increased infectivity and deteriorating epidemiological situations. However, the relation between this variant and severity of COVID-19 infection was contradictory. Conclusion: The COVID-19 pandemic continues to spread worldwide. It is necessary to anticipate large clinical cohorts to evaluate the virulence and transmissibility of SARS-CoV-2 mutants.

## 1. Introduction

At the end of 2019, an epidemic of severe respiratory infections and pneumonia (named COVID-19) began in Wuhan, China. The cause of this outbreak is the severe acute respiratory syndrome coronavirus 2 (SARS-CoV-2) virus. The disease is highly contagious, and the spread of COVID-19 has been taking place at varying rates globally. The World Health Organization (WHO) declared it a Public Health Emergency of International Concern on 30 January 2020 and then a global pandemic on 11 March 2020, less than three months after its appearance [1] (https://www.who.int/emergencies/diseases/novel-coronavirus-2019/events-as-they-happen (accessed on 26 April 2021)). This pandemic is the cause of an unprecedented health care crisis worldwide, with more than 110 million confirmed cases and more than 2,400,000 caused deaths to date [2].

Differences in severity have been observed with respiratory viruses, including influenza viruses, rhinoviruses, and coronaviruses [3,4,5]. SARS-CoV-2 primarily affects the respiratory system and the severity of the disease ranges from asymptomatic infection to severe acute respiratory distress [6]. Additionally, neurological symptoms including, notably, anosmia and ageusia are frequent [6], and some patients may also present with cutaneous [7] and gastrointestinal symptoms [7]. Finally, thrombotic, and thromboembolic diseases appeared to be frequent complications in COVID-19 patients [7]. As a consequence, the severity of the disease may greatly vary depending on the clinical presentation and the organs affected by the disease. In addition, the severity of the disease and the mortality rate are related to many host factors, including age, gender, chronic conditions, comorbidities, race, and ethnicity [6].

On the other hand, the virus mutation is also thought to affect the severity of the disease [8,9,10,11]. Since the emergence of severe acute respiratory syndrome coronavirus 2 (SARS-CoV-2), genetic variants have been identified. In particular, the recent identification of new variants of concerns (VOC) in the UK (named 20I/501Y.V1 or B.1.1.7), South Africa (20H/501Y.V2 or B.1.351), Brazil (P1) and California, USA (B.1.427 and B.1.429), is causing concerns (Table 1) [12,13]. In a study conducted on more than 10,000 SARS-CoV-2 genomes from four databases from patients in 68 countries, 5775 distinct genomes were identified, including 2969 missense mutations and 36 stop-gained variants [14]. Investigation of a possible selective advantage or of an association with clinical severity of these variants is of paramount importance. Mutations in the gene encoding Spike protein of SARS-CoV-2 have been showed to affect both the virus infectivity and antigenicity in vitro [15]. In Marseille, France, a two-act pattern of incidence of COVID-19 cases occurred and significant differences in clinical outcomes were observed between patients seen in March–April 2020 and those seen in June–August [16,17].

We aim to conduct this review to summarize the relation between viral mutation and infectivity of SARS-CoV-2 and viral mutation and severity of COVID-19 in vivo and in vitro.

## 2. Materials and Methods

### 2.1. Search Strategy and Selection Criteria

This review was conducted according to the Preferred Reporting Items for Systematic Reviews and MetaAnalyses (PRISMA) guidelines (http://www.prisma-statement.org (accessed on 31 May 2021)). The following databases were investigated in an attempt to identify all relevant studies published on: PubMed (http://www.ncbi.nlm.nih.gov/pubmed (accessed on 31 May 2021)), Web of Science (https://www.webofknowledge.com/) (accessed on 31 May 2021) and Google Scholar (http://scholar.google.fr/) (accessed on 31 May 2021). The most recent search was conducted on 31 May 2021. The topic search terms used for searching through the databases were the following:

#1: “variant” OR “variation” OR “mutation”;

#2: “SARS-CoV-2” OR “COVID-19”;

#3: “severity” OR infectivity”;

#4: #1 AND #2 AND #3.

Only articles published in English were included. For inclusion, articles had to fulfill two criteria: (1) be related to variants of SARS-CoV-2 virus and (2) describe the relation between viral mutation and infectivity of SARS-CoV-2 or severity of COVID-19. Reference lists of selected articles were screened to identify studies that might have been missing from the research.

After manually removing duplicates, three researchers (TLD, HVT and GP) independently performed the screening of the abstracts, applying the inclusion and exclusion criteria. In addition, articles without an abstract were included for full-text screening and assessed at this stage. Any discordant results were discussed in a consensus meeting. After screening the abstracts, the full texts of the articles were assessed for eligibility by the same three researchers and were selected or rejected for inclusion in the systematic review.

### 2.2. Data Collection Process

The following data (if available) were extracted from each article: country where patients were sampled, time period of the study, number of patients, type of clinical sample, genomic methods, characteristics of variants and outcome.

### 2.3. Data Synthesis and Analysis

As a result of the nature of the studies and the heterogeneity in patient populations, a formal meta-analysis was not possible. Therefore, the study results were summarized to describe the relation between genetic variation in SARS-CoV-2 and viral infectivity or severity of COVID-19. When possible, percentages not presented in the articles were calculated from the available data.

## 3. Results

### 3.1. Study Selection and Types of Studies

The study selection is presented in the flow diagram (Figure 1). The search algorithm produced 1642 articles from the PubMed, Web of Science and Google Scholar databases. After removing duplicates, 758 articles were scanned, based on their titles and abstracts. A total of 254 articles were processed for full-text screening. Sixty-three articles met the inclusion criteria and were included in the qualitative synthesis of the systematic review (Figure 1) [8,9,10,11,15,16,17,18,19,20,21,22,23,24,25,26,27,28,29,30,31,32,33,34,35,36,37,38,39,40,41,42,43,44,45,46,47,48,49,50,51,52,53,54,55,56,57,58,59,60,61,62,63,64,65,66,67,68,69,70,71,72,73].

Of the 63 publications included, five were non-peer reviewed preprints [16,17,52,54,66]. Eleven articles reported in vitro, in silico or animal model studies [15,18,19,22,23,24,25,26,27,28,29]; nineteen articles reported clinical studies [16,37,38,39,41,42,43,44,45,46,47,48,49,51,70,71,72,73]. The remaining 33 articles analyzed SARS-CoV-2 genomes downloaded from the GISAID or other available databases with patient status [8,9,10,11,20,21,30,31,32,33,34,35,36,40,50,52,53,54,55,56,57,58,59,60,61,62,63,64,65,66,67,68,69].

Most clinical studies were conducted before July 2020, corresponding to the first phase of the COVID-19 pandemic. Only four studies were conducted during two different episodes of the pandemic (from March to May and May to July 2020 in one study, from March to November in two studies and from March 2020 to January 2021 in one study) [41,46,70]. Most of the studies were conducted in the UK (9), followed by France (6), the USA (6), China (2), Singapore (2), Brazil (1), UAE (1), Vietnam (1) and Uruguay (1).

### 3.2. Relation between Viral Mutation and Infectivity of SARS-CoV-2

Twenty-five studies reported the relation between viral mutations and virus load or infectivity of SARS-CoV-2 (Table 2), of which five articles (two in vitro/animal model studies [18,19], two studies based on SARS-CoV-2 genomes downloaded from electronic databases or community-based testing datasets (bigdata analysis) [20,21] and one clinical study [16]) showed no impact of mutations with infectivity of SARS-CoV-2. The remaining 20 studies (nine in vitro/animal model or in silico studies [15,22,23,24,25,26,27,28,29], eight bigdata analyses [9,30,31,32,33,34,35,36] and three clinical studies [37,38,39]) showed that SARS-CoV-2 variants were significantly associated with increase in infectivity. All studies performed on the variants of concern (VOCs) have linked these mutations to increased viral transmissibility [23,27,28,29,34,35,36,38,39], except for one study by Li et al. [19]. In this in vitro study, the authors showed a non-significant increase in infectivity in cell lines for any of pseudotyped viruses with the B.1.351 variant compared to the D614G variant [19]. 

### 3.3. Relation between Viral Mutations and Severity of COVID-19 Infection

A total of 45 articles addressed the effect of viral mutations on severity of COVID-19 in patients, of which fifteen studies (five bigdata analyses [21,30,33,34,40] and ten clinical studies [39,41,42,43,44,45,46,47,48,49]) showed no association between SARS-CoV-2 variants and severity of disease. Three studies (one bigdata analysis [50] and two clinical studies [17,51]) reported that the SARS-CoV-2 variants were significantly associated with decrease in severity of disease. The remaining 27 studies (one animal study [18], 22 bigdata analyses [8,9,10,11,52,53,54,55,56,57,58,59,60,61,62,63,64,65,66,67,68,69] and four clinical studies [70,71,72,73]) described the positive association between the different variants with severity of COVID-19 (Table 3).

A clinical study by Long et al. showed that infection with SARS-CoV-2 variants harboring the D614G substitution was not associated with disease severity, overall mortality, transfer to ICU, mechanical ventilation and length of stay at hospital [41]. This result was supported by other research [33,45]. In a study conducted among 44 Vietnamese patients, 85 mutations covering 67 variant types were reported, of which P323L and D614G variants were the most frequent (present in 40/44 patients), followed by C241U (39/44) and GGG to AAC at 28881-3 variants (33/44). However, these mutations were not significantly associated with phenotype of illness [44]. Genomic investigation of 309 SARS-CoV-2 isolates obtained from patients seen in Marseille in March–April revealed specific mutations clustering in five main groups with no marked correlation with clinical severity of the disease [16]. Additionally, no significant difference in rate of transfer to ICU or mortality was identified between patients infected with the Marseille-1 or Marseille-4 variants (during the second episode of the pandemic) and those with clade 20A strains that predominantly circulated during the first phase of the pandemic in Marseille [46,47]. A study by Zhang et al. conducted in China showed no significant differences between two variants (clade I (ORF3a: p.251G > V, or S: p.614D > G (subclade G)); clade II (ORF8: p.84L > S (28,144U > C) and ORF1ab: p.2839S (8782C > U)) regarding disease severity and blood parameters indicative of severity [24]. No significant difference in clinical presentation was observed between hospitalized patients harboring or not harboring the D26 and D34 variants in the ORF6 protein [48]. A study conducted on isolates from patients in Washington, US, allowed the identification of two major clades distinguished by twelve polymorphisms in five genes. No significant difference regarding mortality was observed among patients infected with these two clades [43]. In a study conducted on 88 patients in the USA, most of the sequences (93%) were clustered in three main clades (clades 1, 2 and 3), defining mutations at the US level. The authors showed that the viral mutations have had no effect on time to symptom onset or disease severity [37]. Pawan et al. identified seven different variants from 3068 SARS-CoV-2 genomes obtained from GISAID, of which three clades (V, S and GH) were of no effect on the outcome of patients [54]. The new variant N501Y has been described as being more transmissible but is not associated with severity of COVID-19 infection [34].

A negative relation between severity of COVID-19 infection and virus mutations was described in several studies. In Singapore, a SARS-CoV-2 variant with a 382-nucleotide deletion (Δ382) that eliminates open-reading frame (ORF) 8 transcription was detected in a cluster of cases in January and February, 2020, and was associated with less severe infection in patients, in terms of hypoxia requiring supplemental oxygen [51]. A French study conducted during the second phase of the epidemic in the country showed that SARS-CoV-2 mutation rate was negatively associated with mortality rate [17]. Additionally, the Marseille-1 variant was associated with lower frequency of hospitalization than clade 20A strains [46]. In addition, the 11,083G > U mutation was significantly associated with asymptomatic patients [50].

By contrast, ORF1ab 4715L and S protein 614G variants were significantly more frequent in patients from countries where high fatality rates were also reported [8,10,11,57,59,60]. Patients infected with virus clades L, G and O were also exposed to a higher risk of severe infection than the base level [54]. Mutations at NSP6 and S proteins show a tendency to increase the death rate [56]. Majumda et al. analyzed 218 viral strains obtained from 15 countries. Their result showed that mutation in ORF3 protein increased the mortality of COVID-19 infection [9]. Among 1096 SARS-CoV-2 complete sequences downloaded from UK Biobank, 216 different verified super-variants were identified with eight predominant generic variants (chr6_148, chr7_23, chr2_197, chr2_221, chr8_99, chr10_57, chr16_4 and chr17_26). These variants were significantly associated with an increase in COVID-19 mortality [55]. By analyzing the data from datasets provided by Public Health England, Davies et al. showed that the 501Y variant was associated with a 12–64% higher risk of death [63].

Stauft et al. conducted an animal model study to evaluate the pathogenicity of D614 and G614 variants in the Spike protein. The authors showed that there was no difference in pathology in lung tissues of hamsters infected with these variants [18].

A total of ten studies reported the relation between new VOCs and severity of COVID-19, of which three studies (two bigdata analyses and one clinical study) showed negative associations. Most studies (6/7) describing a positive association between these variants and severity of infection used bigdata analyses. Only one clinical study [73] conducted using 68 patients showed a higher hospitalization rate in patients infected with the P.1 variant.

## 4. Discussion

The SARS-CoV-2 virus, due to the lack of proofreading activity of the RNA-dependent RNA polymerase, has high mutation rates that may have important effects on the pathogenicity and transmissibility of the virus [16]. The identification of genome variations of SARS-CoV-2 and their relationships with viral infectivity or severity of COVID-19 is therefore important for controlling and surveying the evolution of the pandemic [14,74]. In addition, mutation rate of SARS-CoV-2 determines the evolution of this virus and the risk of emergent infectious diseases [74]. In a study by Koyama et al., the median mutation rate of SARS-CoV-2 was estimated at 1.12.10^−3^ mutations per site/year (95%CI = [9.86.10^−4^–1.85.10^−4^]) [74]. A high mutation rate of around 30% was observed among 95 full-length genomic sequences [75]. An analysis of 48,635 samples showed an average of 7.23 mutations per sample [76]. To date, 46,251 SARS-CoV-2 mutations were documented in the public databases (http://cov-glue.cvr.gla.ac.uk/#/replacement (accessed on 31 May 2021)). Numbers of variations are the highest in the NSP3 protein, followed by S protein, NSP12 protein, NSP2 protein, NSP 13 protein, NSP14, and NSP4 protein. By contrast, very little divergence was documented in NSP11, ORF10, ORF7b and E proteins [74] (http://cov-glue.cvr.gla.ac.uk/#/replacement (accessed on 31 May 2021)). Figure 2 shows the positions of the mutations and deletions in the genome and of amino acid substitutions in the virion.

A key element of the coronavirus host range is determined by the binding affinity between the spike S protein and the cellular receptor. All mutations in the S protein could influence host range and transmissibility of the virus [16]. The SARS-CoV-2 Spike protein is a class I fusion protein that forms trimers on the viral surface: it is heavily glycosylated, which enables entry into host cells [77]. Angiotensin-converting enzyme 2 (ACE2) is the target receptor of the SARS-CoV-2 virus for entry into the host cell [77]. The main effect of the D614G mutation is to increase the availability of spike trimer components in the conformation and permits enhancing the binding of the virus spike to the ACE2 receptor. The in vitro and in vivo studies to date showed that the mutation D614G in Spike protein was associated with higher viral loads and probably with enhanced transmissibility of the virus [22,25,30]. Therefore, this mutation emerged and has become the dominant form in the global pandemic worldwide within a matter of months. It suggests that G614 may have a fitness advantage [78]. The frequency of S protein 614G was significantly associated with high fatality rates in several countries, as reported in studies which analyzed SARS-CoV-2 sequences from GISAID database [8,10,11]. However, clinical studies showed that this mutation did not correlate with severity of COVID-19, including mortality, transfer to ICU, mechanic ventilation, or length of stay at hospital [30,37,41,42,44,45].

In addition, clinical studies have shown that other viral mutations were not related to severity of COVID-19 infection or were associated with less severe infection in patients. Young et al. showed that the patients infected with Δ382 had lower concentrations of inflammatory biomarkers. Furthermore, these patients had a higher concentration of SDF-1 α, which is low in patients with hypoxemia [51]. Interestingly, the replication capacity of ∆382 variant in vitro is similar to that of wild-type SARS-CoV-2. It suggests that this mutation does not reduce replicative fitness [51]. In a study by Colson et al., the authors demonstrated seven new mutations of SARS-CoV-2, named “Marseille-1” to “Marseille-7”. Moreover, heterogeneity of the sequences produced from June to August 2020 (second outbreak) was higher than in sequences produced from February to May 2020 (first outbreak) (7.6.10^−4^ ± 3.8.10^−4^ versus 2.3.10^−4^ ± 1.1.10^−4^). This result indicates that the rate of virus mutation has increased rapidly. By contrast, the mortality of COVID-19 patients during the second outbreak was lower than that of those in the first one [17].

Recently, 501Y1, 501Y2 and P1 variant emergence in UK, South Africa and Brazil and the successive spread beyond the country of origin [79] have led to significant concern by medical and political authorities in many countries with extensive media coverage. The current evidence may suggest that these variants have led to increased infectivity and deteriorating epidemiological situations [34,79,80]. However, the relation between these variants, particularly the 501Y variant, and severity of COVID-19 is contradictory [34,63,79,80]. In our analysis, 8/10 studies reported a positive association between these VOCs and severity of COVID-19. Almost all of these studies were bigdata analyses. In fact, in one clinical study conducted using 496 patients, with 341 having samples that could be sequenced, no evidence of an association between severity of disease and death and B.1.1.7 variant was observed [39]. In addition, Dao et al. showed a lower rate of hospitalization associated with N501Y variants as compared to Clade 20A and the Marseille-4 variant [70]. To date, only one clinical study reported a higher hospitalization rate in 68 patients infected with the P.1 variant [73]. These contradictory results can be explained by the much faster rate of spread of VOCs compared to that of the original virus strain or previous variants. This means that when the number of infected persons in the population increases, the absolute number of severe cases and deaths may also increase. Therefore, the studies based on the community-based testing dataset reported an increase in mortality in patients infected with VOCs. The absolute risk increase affecting individual patients is likely minimal.

Our study has some limitations. We screened papers published only in English and reported in PubMed, Web of Science and Google Scholar. Ongoing research projects have not been used. For example, in our University Hospital Institute, a large cohort study aiming at comparing the demographic and clinical characteristics of patients infected with several new variants of SARS-CoV-2 virus during July to September 2020 is ongoing. In addition, a new variant, L18F, has been recently reported with 1186 spike L18F VOC genomes in the UK. The weekly growth rate of the L18F increased 1.75-fold, compared with the VOC genomes non-mutated at residue 18 [81]. Additionally, the VOC P1 was significantly associated with lower Ct values and with an increase in positive samples from 0 to 87% in Manaus, Brazil, between 2 November 2020, and 4 January 2021, suggesting high transmissibility [38]. However, the number of infections with these variants remains too small to perform an analysis on their infectivity [38,81]. Nevertheless, our review gives an overview on the relation between SARS-CoV-2 genetic variations and viral infectivity or severity of COVID-19 infection. In conclusion, most studies showed that some genetic variants of the virus were associated with high virus load. However, to date, the evidence of the association between viral mutation and severity of the disease is scant. On the other hand, severity and outcome of COVID-19 infection depend on the host’s genetic factors, on the treatment and clinical management, which have been improved, and on increased hospital capacity and response speed. The COVID-19 pandemic continues to spread worldwide. It is necessary to anticipate large clinical cohorts to evaluate the virulence and infectivity of SARS-CoV-2 mutants.

## Figures and Tables

**Figure 1 jcm-10-02635-f001:**
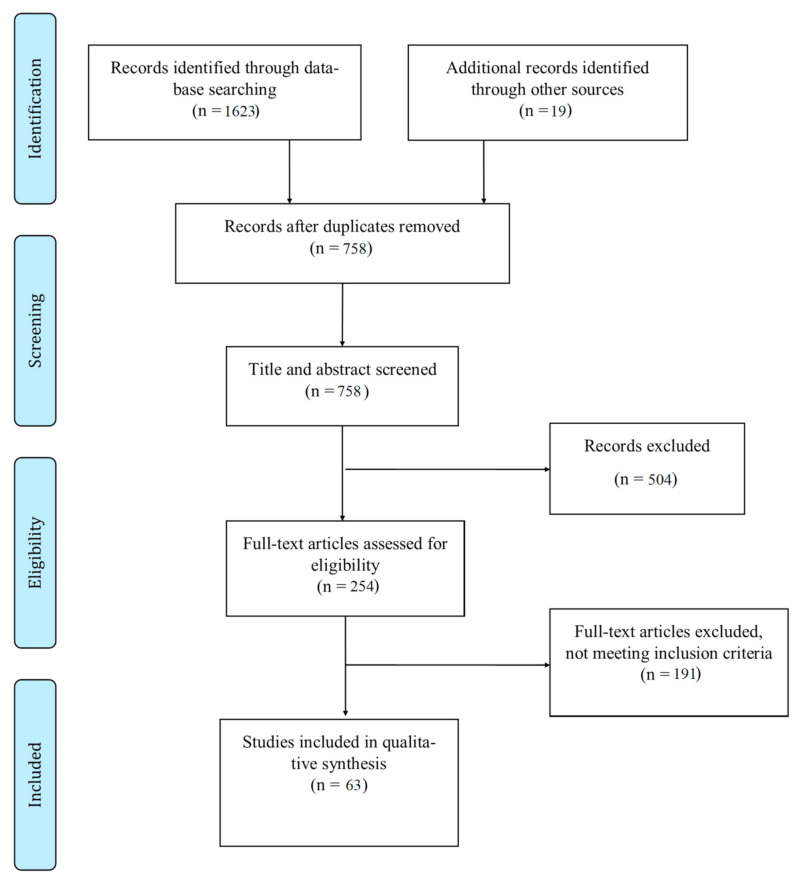
Study flow chart.

**Figure 2 jcm-10-02635-f002:**
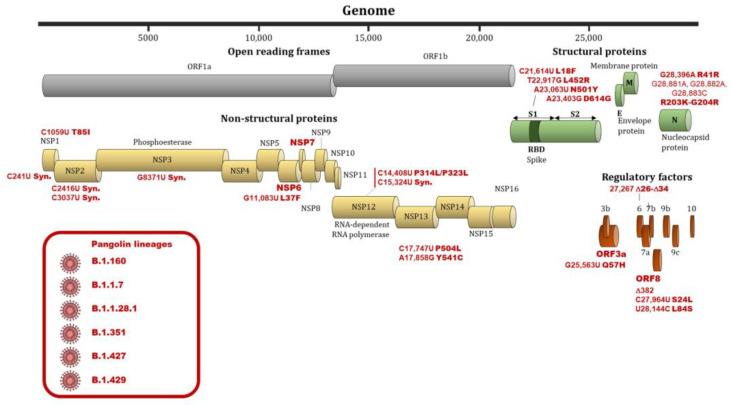
Nucleotide and amino substitutions or deletions in the SARS-CoV-2 genome and viral clades associated with differences regarding infectivity and clinical severity. Footnote: Substitutions, deletions and genes involved in differences regarding infectivity and clinical severity are indicated by a red bold font. Viral clades involved in differences regarding infectivity and clinical severity are indicated by a red font and are shown in a red frame.

**Table 1 jcm-10-02635-t001:** Description of variants of concern (VOCs), up to 31 May 2021.

Lineage	Country First Detected (Community)	Mutation/Deletion	Year and Month First Detected	Number of Affected Countries *
B.1.1.7	United Kingdom	69del, 70del, 144del, (E484K), (S494P), N501Y, A570D, D614G, P681H, T716I, S982A, D1118H (K1191N)	September 2020	136
B.1.351	South Africa	D80A, D215G, 241del, 242del, 243del, K417N, E484K, N501Y, D614G, A701V	September 2020	92
P.1	Brazil	L18F, T20N, P26S, D138Y, R190S, K417T, E484K, N501Y, D614G, H655Y, T1027I	December 2020	54
B.1.617.2 ^¥^	India	T19R, (G142D), 156del, 157del, R158G, L452R, T478K, D614G, P681R, D950N	December 2020	57
B.1.427 ^⁋^	United States (California)	L452R, D614G	January 2021	−
B.1.429 ^⁋^	United States (California)	S13I, W152C, L452R, D614G	January 2021	−

* As of 31 May 2021 (https://covariants.org/ (accessed on 31 May 2020)). ^¥^ According to European Centre for Disease Prevention and Control. ^⁋^ According to American Centre for Disease Prevention and Control.

**Table 2 jcm-10-02635-t002:** Viral mutation and infectivity of SARS-CoV-2.

Ref.	Country Where Patients Were Sampled	Period of Time	Number of Patients	Type of Samples	Sequencing Methods/Data Availability	Variants	Outcome
**1.1. Studies showing no impact**							
1.1.1. In vitro and/or animal model studies							
[18]	NA	NA	Animal model study with a mutant virus and a wild-type virus	NA	NA	SARS-CoV2 D614 and G614 variants in the Spike glycoprotein	Hamsters infected with the two variants exhibited comparable viral loads in lung tissues as well as similar amounts of virus shedding in nasal washes
[19]	NA	NA	In vitro study	NA	NA	501Y.V2 variant	No significant increase in infectivity was observed in these cell lines for any of the pseudotyped viruses with 501Y.V2-related mutations compared to the reference 614G variant
1.1.2. Studies based on SARS-CoV-2 genomes downloaded from electronic databases or based on community-based testing dataset							
[20]	Various		46,723	ND	46,723 complete SARS-CoV-2 genomes downloaded from GISAID	12,706 variable positions	None of the recurrent SARS-CoV-2 mutations were associated with increased viral transmission
[21]	Singapore	22 January to 22 April 2020	The first 10,000 COVID-19 cases were extracted from the Ministry of Health database. 319 patients had SARS-CoV-2 sequences available	Nasopharyngeal swabs	Sequencing of SARS-CoV-2 was performed in one of four laboratories in Singapore together with GISAID submission and case matching. Pangolin COVID-19 Lineage Assigner and CoVsurver were used to assign lineage and clade to each sequence, respectively	29 were infected with clade S, 90 with clade L/V, 96 with clade G (containing D614G variant), and 104 with other clades ‘O’	No significant difference in the transmissibility of clade G infections was observed
1.1.3. Clinical studies							
[16]	France	29 February to 4 April 2020	309	Nasopharyngeal swabs	Sequencing by Illumina protocols on MiSeq platform (Illumina)	A total of 321 mutational events were reported in the SARS-CoV-2 genomes divided into 5 clusters. Cluster 1 (44 patients, 14.2%, positions (28,881-28,882-28,883)) with two non-synonymous mutations in protein N (R203K; G204R). Cluster 2 (39 patients, 12.6%, position 15,324) contains a synonymous mutation (C15324U). Cluster 3 (126, 100 and 211 patients, at positions 2416, 8371, 25,563, respectively) includes one synonymous mutation (C2416U), and two non-synonymous mutations (nsp3: Q1884H; ORF3a: Q57H). Cluster 4 (68 patients, 22%, position 1059) contains one non-synonymous mutation (nsp2: T85I). Finally, cluster 5 (from 297 to 303 patients, 96–98%, positions 241, 3037, 14,408, 23,403) displays one mutation in 5′UTR (C241U), one synonymous mutation (C3037U) and two non-synonymous mutations (nsp12b: P314L, S protein 133 D614G)	Coronavirus genome isolates from 38 patients’ isolates with PVirO were widely distributed across the groups, including diverse mutational events meaning that there is no correlation between higher viral loads
**1.2. Studies showing increased infectivity**							
1.2.1. In vitro—In silico and/or animal model studies							
[15]	NA	NA	In vitro study	NA	NA	S mutants reported in the public domain or mutants at putative N-linked glycosylation sites	Pseudotyped viruses expressing either the D614G single mutation or a combination of mutations that included D614G are more infectious than the reference strain, whereas no difference was found between single D614G and D614G combination variants
[22]	NA	NA	In vitro study	NA	NA	SARS-CoV2 D614G mutation in the Spike glycoprotein	Pseudovirus G614 infected hACE2-293T cells with approximately 9-fold higher efficiency than did Pseudovirus D614
[23]	NA	NA	In vitro study	NA	NA	New 501Y variant	501Y variant binds to human Angiotensin Converting Enzyme 2 (ACE2) approximetely10 times more tightly than the native version
[24]	NA	NA	In vitro and animal model study	NA	NA	SARS-CoV2 D614G mutation in the Spike glycoprotein	D614G mutation increases the infectivity of SARS-CoV-2 produced from a human lung cell line. Hamsters infected with the G614 variant produced higher infectious titers in the nasal washes and trachea, but not lungs, confirming clinical evidence that the D614G mutation enhances viral loads in the upper respiratory tract of COVID-19 patients
[25]	NA	NA	In vitro and animal model study with a mutant virus and a wild-type virus	NA	NA	SARS-CoV-2 D614G mutation in the Spike glycoprotein	SARS-CoV-2 variants harboring the D614G substitution replicated more efficiently in some immortalized epithelial cell lines and exhibited significantly faster droplet transmission between infected hamsters than the wild-type virus
[26]	China	22 January to 4 February 2020	In vitro and clinical study among 11 patients	Sputum, stool and nasopharyngeal swabs	Deep sequencing by the Novaseq 6000 platform (Illumina)	33 mutations were identified in 11 isolates	Different viral isolates exhibit significant variations of viral load when infecting Vero-E6 cells. ZJU-1, which clusters with the S-D614G clade, has a viral load 19 times higher than ZJU-2 and ZJU-8. A near 270-fold difference in viral load was observed between ZJU-10 and ZJU-2 at 24 h post infection. In addition, there was a higher viral load leads to a higher cell death ratio
[27]	USA	1 September 2020 to 29 January 2021	In vivo and in vitro study	Nasopharyngeal swabs	2172 SARS-CoV-2 genomes were sequenced	B.1.427/B.1.429 and spike L452R variants	B.1.427/B.1.429 variant was 20% more transmissible with 2-fold increased shedding in vivo. The spike L452R mutation conferred increased infectivity in vitro
[28]	−	−	In silico study	−	Computational methods, including Molecular Operating Environment (MOE) analysis and software, were used to predict the outcome of substitutions with regard to the protein structure to examine the features acquired by the new variants that enable them to increase the rate of infection and spreading without increasing the severity of COVID-19, the pathology resulting from the infection	B.1.1.7 and B.1.351 variants	B.1.1.7 and B.1.351 variants acquired an increased transmissibility
[29]	NA	NA	In vitro study	NA	NA	B.1.351 variant	The three different pseudoviruses of B.1.351 lineage had significantly increased infectivity compared with other mutants that indicated Wuhan strains
1.2.2. Studies analyzing SARS-CoV-2 genomes downloaded from electronic databases or based on community-based testing dataset							
[9]	23 countries		Approximately 20,000 case reports	ND	SARS-CoV-2 strains for each country were extracted from NextStrain open-source project. Amino acid sequences of ORF3a protein were downloaded from NCBI protein database	218 viral strains from 15 countries were further analyzed for amino acid mutations from NextStrain database	Mutation in ORF3a protein was associated with increased infection of SARS-CoV-2
[30]	UK	March to May 2020	999	Throat or combined nose/throat swabs	Long-read whole genomesequencing (Oxford Nanopore Technologies (ONT), Oxford, UK) using the ARTIC network protocol	SARS-CoV2 D614G mutation in the Spike glycoprotein	SARS-CoV-2 variants harboring the D614G substitution were associated with potentially higher viral loads in COVID-19 patients
[31]	USA		7823		28,726 complete SARS-CoV-2 genome sequences downloaded from GISAID	4968 single mutations were detected with the top eight missense mutations (i.e., 14,408C > U-(P323L), 23,403A > G-(D614G), 25,563G > U-(Q57H), 1059C > U-(T85I), 28,144U > C-(L84S), 17,858A > G-(Y541C), 17,747C > U-(P504L), and 27,964C > U-(S24L)) identified	Based on co-mutation and time evolution analysis, three concurrent mutations 17,747C > U-(P504L), 17,858A > G-(Y541C), and 28,144U > C tend to fade out, while the other five concurrent mutations can enhance the infectivity of SARS-CoV-2
[32]	17 countries		24,175	ND	24,175 complete SARS-CoV-2 genomes downloaded from GISAID	11,904 single mutations found in 6 distinct clusters	Mutations on the RBD strengthen the binding of S protein and ACE2, leading to more infectious SARS-CoV-2
[33]	UK	29 January to 16 June 2020	ND	ND	21,231 614G and 5755 614D de-duplicated whole-genome sequences were downloaded from The COVID-19 Genomics UK consortium dataset	245 and 62 clusters of 614G and 614D variants containing UK virus genomes from 10 or more different patients were identified, respectively	614G variant was associated with higher viral load and younger age of patient.
[34]	United Kingdom	−	−	−	Bigdata analysis from COVID-19 Genomes UK dataset	VOC 202012/01 variant	VOC 202012/01 variant was 43–82% more transmissible than pre-existing variants
[35]	UK	1 August to 31 December 2020	87,830	−	94,934 sequences originating from UK were downloaded from GISAID database. The surveillance data of daily number of COVID-19 cases in UK were collected from the World Health Organization (WHO) COVID-19 surveillance platform	VOC 202012/01 variant	VOC 202012/01 variant associated with 46–58% increase in infectivity (reproduction number)
[36]	UK	16 November to 13 December 2020	143,994 samples obtained from staff and residents of long-term care facilities throughout England	−	B.1.1.7 variant was identified in samples with S gene target failure on PCR	B.1.1.7 and non-B.1.1.7 variants	B.1.1.7 variant was significantly associated with decrease in median Ct values
1.2.3. Clinical studies							
[37]	USA	Mid-March 2020	88	Nasopharyngeal swabs	Library sequencing performed on the Nanopore MinION device using FLO-MIN106D Type R9.4.1 flow cells	Most of the sequences (93%) clustered in three main clades (Clade 1, 2 and 3), defining mutations at the US level	Patients infected with Clade 1 viruses had significantly higher average viral loads in their upper airways relative to patients infected with Clade 2 viruses
[38]	Brazil	November to December 2020	147 samples	−	Sequencing was conducted using ARTIC V3 multiplexed amplicon scheme and the MinION sequencing platform	P.1 variant	P.1 variant was associated with increased transmissibility
[39]	UK	09 November 2020 to 20 December 2020	496 with 341 patients had sequenced samples	Nose and throat samples	Illumina MiSeq 500v2 kits and MiSeq reagent cartridge V2 were used for sequencing. The COG-UK Mutation Explorer was used to identify potential mutations of concern	B.1.1.7 variant (198 patients) and non-B.1.1.7 variant (143 patients)	Significantly lower Ct values associated with B.1.1.7 compared with non-B.1.1.7

**Table 3 jcm-10-02635-t003:** Viral mutation and severity of COVID-19.

Ref.	Country Where Patients Were Sampled	Period of Time	Number of Patients	Type of Samples	Sequencing Methods/Data Availability	Variants	Outcome
**2.1. Studies showing no impact**							
2.1.1. Studies analyzing SARS-CoV-2 genomes downloaded from electronic databases or based on community-based testing dataset							
[40]	−	1 September 2020 to 31 January 2021	182,982	−	182,982 complete SARS-CoV-2 strains were retrieved from GISAID	N501Y variant	No statistically significant evidence of change in COVID-19 mortality risk associated with 501Y variants was observed
[21]	Singapore	22 January to 22 April 2020	The first 10,000 COVID-19 cases were extracted from the Ministry of Health database. 319 patients had SARS-CoV-2 sequences available	Nasopharyngeal swabs	Sequencing of SARS-CoV-2 was performed in one of four laboratories in Singapore together with GISAID submission and case matching. Pangolin COVID-19 Lineage Assigner and CoVsurver were used to assign lineage and clade to each sequence, respectively	29 were infected with clade S, 90 with clade L/V, 96 with clade G (containing D614G variant), and 104 with other clades, “O”	Infections with clade S or clade O were associated with lower odds of developing hypoxia requiring supplemental oxygen compared with clade L/V. No significant difference in the severity of clade G infections was observed
[30]	UK	March to May 2020	999	Throat or combined nose/throat swabs	Long-read whole genome sequencing (Oxford Nanopore Technologies (ONT), Oxford, UK) using the ARTIC network protocol	SARS-CoV2 D614G mutation in the Spike glycoprotein	SARS-CoV-2 variants harboring the D614G substitution were not associated with disease severity
[33]	UK	29 January to 16 June 2020	ND	ND	21,231 614G and 5755 614D de-duplicated whole-genome sequences were downloaded from The COVID-19 Genomics UK consortium dataset	245 and 62 clusters of 614G and 614D variants containing UK virus genomes from 10 or more different patients were identified, respectively	614G variant was not associated with mortality or severity of COVID-19
[34]	United Kingdom	−	−	−	Bigdata analysis from COVID-19 Genomes UK dataset	VOC 202012/01 variant	No difference in severity of disease (hospitalization, transfer to ICU and death) was observed, compared to VOC 202012/01 and pre-existing variants
2.1.2. Clinical studies							
[41]	USA	5 March to 11 May 2020 (first wave) and 12 May to 7 July 2020 (second wave)	1026 (first wave) and 4059 (second wave)	Nasopharyngeal swabs	Long reads were generated with the LSK-109 sequencing kit, 24 native barcodes (NBD104 and NBD114 kits), and a GridION instrument (Oxford Nanopore). Short reads were generated with a NexteraXT kit and a NextSeq 550 instrument (Illumina)	SARS-CoV2 D614G mutation in the Spike glycoprotein	No relationship between virus clades and disease severity (overall mortality, transfer to ICU, mechanical ventilation and length of stay)
[42]	China	20 January–25 February 2020	112	Sputum or nasopharyngeal swabs	Sequencing by Illumina protocols on MiSeq platform (Illumina)	Clade I (ORF3a: p.251G > V (subclade V), or S: p.614D > G (subclade G)). Clade II (ORF8: p.84L > S (28,144U > C) and ORF1ab: p.2839S (8782C > U)	There were no significant differences between variants regarding disease severity, leukocytes, lymphocyte and platelet count, CD3 T cell count, hemoglobin, C-reactive protein, Lactose dehydrogenase, complement C3,D-dimer or IL-6 and IL-8 level, or the duration of virus shedding after onset
[43]	USA	1 March to 15 April 2020	190	Nasopharyngeal samples	Samples were sequenced on MiSeq, NextSeq or NovaSeq instruments (Illumina) using 1 × 185, 1 × 75, or 1 × 100 runs, respectively	97 samples corresponded to what we refer to as “Clade 1” and 91 corresponded to “Clade 2”. Two of 190 samples did not fall into either of the two major clades. When mapped onto GISAID and NextStrain clades, in clade 1, 89 corresponded to clades GH/20C, 6 were mapped to G/20A, and 2 were mapped to G/20B. In clade 2, 86 corresponded to S/19B, and 5 were mapped onto L/19A. 2 of the 190 samples that did not fall into either of the major clades corresponded to GH/20C and S/19B	A trend toward higher rates of hospitalization of patients with Clade 2 virus was observed (*p* = 0.06). Mortality was not significantly different in patients infected with Clade 1 and 2 viruses
[44]	Vietnam	6 March to 15 April 2020	44	Nasopharyngeal and oropharyngeal swabs	Sequencing was performed on an Illumina Miseq platform (Nextera XT Library preparation kit)	85 mutations covering 67 variant types among the 44 SARS-CoV-2 genomes. The most ubiquitous modifications were C3037U, C14,408U (P323L) and A23,403G (D614G) occurring in 40/44 samples. Two other variants C241U and GGG to AAC at 28881-3 were detected in 39 and 33 sequences, respectively	These mutations were not associated with differences in phenotype of illness
[45]	Uruguay	March to May 2020	44	Naso-oropharyngeal swabs	Whole SARS-CoV-2 genomes were sequenced using Illumina NovaSeq 6000	D614G mutation	The spike D614G mutation and clade G-related viruses were not associated with any clinical parameters, severity, or lethality of COVID-19 infection
[46]	France	March to 14 August 2020	417	Nasopharyngeal swabs	Positive samples for SARS-CoV-2 with Ct < 30 were sequenced using next-generation sequencing and Illumina technology. Positive samples with Ct > 30 or with Ct < 30 but genome sequences that were not obtained were tested for Marseille 1 variant by RT-qPCR	Marseille-1 variant and Clade 20A strains	Compared to Clade 20A strains, the Marseille-1 variant was associated with lower frequency of dyspnea, hypoxemia and hospitalization. However, no significant differences of transfer to ICU and death frequency were observed
[47]	France	March to November 2020	759	Nasopharyngeal swabs	Positive samples for SARS-CoV-2 with Ct < 30 were sequenced using next-generation sequencing and Illumina technology	Marseille-4 variant and Clade 20A strains	Compared to Clade 20A strains, the Marseille-4 variant was associated with a lower frequency of cough, rhinitis and olfactory and gustatory disorders but with a higher frequency of hypoxemia. No significant differences of hospitalization, transfer to ICU and death frequency were observed
[48]	France	February to April 2020	229	Nasopharyngeal swabs	Positive samples for SARS-CoV-2 with Ct < 20 were sequenced using an RNA metagenomic next-generation sequencing on an Illumina NextSeqTM 550 with mid-output 2 × 150 flow cell	Two frameshifting deletions were detected in the ORF6 protein at the same position (27,267): D26 and D34	No significant difference in clinical presentation could be observed between hospitalized patients harboring or not harboring the ORF6 deletion
[39]	UK	9 November 2020 to 20 December 2020	496 with 341 patients had sequenced samples	Nose and throat samples	Illumina MiSeq 500v2 kits and MiSeq reagent cartridge V2 were used for sequencing. The COG-UK Mutation Explorer was used to identify potential mutations of concern	B.1.1.7 variant (198 patients) and non-B.1.1.7 variant (143 patients)	No association of the B.1.1.7 variant with severe disease was observed in hospitalized patients
[49]	USA	30 March to 17 July 2020	27 hospitalized patients	Swab eluate or neat endotracheal aspirate or saliva	Genomes were analyzed by reverse transcription of the viral RNA to make a cDNA copy, PCR amplification of genome segments, Nextera library preparation, and Illumina sequencing	All genomes were found to encode the D614G spike polymorphism	No significant associations were found between SARS-CoV-2 variants and clinical outcomes
**2.2. Studies showing a decreased severity of COVID-19**							
2.2.1. Studies analyzing SARS-CoV-2 downloaded from electronic databases or based on community-based testing dataset							
[50]	Various	−	75,775	ND	75,775 SARS-CoV-2 complete genome sequences were downloaded from GISAID database. 9912 samples have patient status information recorded as asymptomatic, symptomatic, hospitalized, intensive care unit, deceased. Of which, 537 samples are labeled as asymptomatic (76) and symptomatic (461) cases	11,083G > U mutation changes leucine to phenylalanine residue at position 37 of NSP6 protein	11,083G > U mutation was significantly associated with asymptomatic patients (OR = 33.4, *p* = 8.45.10^−35^)
2.2.2. Clinical studies							
[17]	France	June to September 2020	691	Nasopharyngeal swabs	Next-generation sequencing using Illumina technology with the Illumina Nextera XT Paired end strategy on a MiSeq instrument	Marseille-1 to Marseille-7, located in most SARS-CoV-2 genes including structural and non-structural genes among which nsp2, nsp3 (predicted phosphoesterase), nsp5 (membrane glycoprotein), nsp12 (RNA-dependent RNA polymerase), S (Spike glycoprotein), ORF3a, E (membrane glycoprotein), M (membrane glycoprotein), ORF8 and N (Nucleocapsid phosphoprotein)	SARS-CoV-2 mutation rate was negatively associated with mortality rate
[51]	Singapore	22 January to 21 March 2020	131	Respiratory sample	2 specific PCRs were used to detect the 382-nucleotide deletion in SARS-CoV-2	92 (70%) were infected with the wild-type virus, ten (8%) had a mix of wild-type and ∆382-variant viruses, and 29 (22%) had only the ∆382 variant	Infection with the ∆382 variant was only associated with lower odds of developing hypoxia requiring supplemental oxygen (adjusted odds ratio 0·07 (95% CI 0·00–0·48)) compared with infection with wild-type virus only
**2.3. Studies showing an increased severity of COVID-19**							
2.3.1. Animal model studies							
[18]	NA	NA	Animal model study with a mutant virus and a wild-type virus	NA	NA	SARS-CoV2 D614 and G614 variants in the Spike glycoprotein	Hamsters infected with the two variants exhibited comparable pathologies in lung tissues
2.3.2. Studies analyzing SARS-CoV-2 downloaded from electronic databases or based on community-based testing dataset							
[8]	50 countries from six geographic areas		12,343	ND	12,343 SARS-CoV-2 sequences isolated in 50 different countries from six geographic areas obtained from GISAID database	1234 mutations, including 57Q > H, 251G > V (ORF3 protein), 265U > I, 378V > I, 5865Y > C, 5828P > L, 4489A > V, 2016U > K, 3606L > F, 4715P > L (ORF1ab protein), 614D > G (S protein), 204G > R, 203R > K, 13P > L (N protein), 175U > M (M protein), 84L > S (ORF8 protein)	ORF1ab 4715L and S protein 614G variants were significantly more frequent in patients from countries where high fatality rates were reported
[9]	23 countries		Approximately 20,000 case reports	ND	SARS-CoV-2 strains for each country were extracted from NextStrain open-source project. Amino acid sequences of ORF3a protein were downloaded from NCBI protein database	218 viral strains from 15 countries were further analyzed for amino acid mutations from NextStrain database	Mutation in ORF3a protein was associated with increased mortality rate of SARS-CoV-2
[10]	Various		ND	ND	SARS-CoV-2 viral spike sequences were accessed from the GISAID database	D614G variant	Both the average and median case fatality rates correlate strongly (*p* < 0.02) with the proportion of G614 variant
[11]	Various		4246	ND	4246 SARS-CoV-2 genomes downloaded from GISAID	D614G variant	D614G variant was associated with high mortality related to COVID-19 in European populations
[52]	Various		152	Not documented	Genomes of SARS-CoV-2 with patient status. Criteria for selection were full-length sequences and high sequencing coverage (downloaded from the GISAID database)	Two genetic variations were observed at the nucleotide position 11,083, namely thymine (11,083U, 75/152 = 49.34%) and guanine (11,083G, 72/152 = 47.37%)	Viruses causing symptomatic cases tended to have 11,083G (N(11,083U)/N(11,083G) = 15/65). The relative risk ratio of developing symptoms given 11,083G to 11,083U was (65/72)/(15/75) = 4.51 times (95% confidence interval = 2.85–7.14), and the odds ratio was estimated to be 37.14 by the Wald method (95% confidence interval = 14.17–97.33)
[53]	Various		73,020	ND	72,331 viral sequences downloaded from GISAID database. Clinical data were available for 5094 patients, and 3184 of them also had follow-up data	2121 different mutations affecting the protein structure were identified	Mutations leading to severe outcome with low prevalence were found in the surface (S) glycoprotein and in NSP7
[54]	Various		3608	ND	3068 SARS-CoV-2 genomes downloaded from GISAID	7 different variants (Clade G, GH, GR, L, O, S and V)	Patients infected with virus clades L, G and O are exposed to higher risk than the base level
[55]	UK	−	1096	ND	1096 SARS-CoV-2 complete sequences were downloaded from UK Biobank	216 different verified super-variants across 10 repetitions of the discovery-validation procedure were found. Two super-variants, chr6_148 and chr7_23, were identified in 4 out of 10 repetitions. Six other super-variants, chr2_197, chr2_221, chr8_99, chr10_57, chr16_4 and chr17_26 were identified in 3 out of 10 repetitions	Eight genetic variants are identified to significantly increased risk of COVID-19 mortality
[56]	Various	April to July 2020	41,304	ND	41,304 SARS-CoV-2 protein sequences from 49 different countries were downloaded from NCBI GenBank	Mutation at NSP6, ORF8, S, M, E and N proteins	A relationship of positive tendency between the death rate and the mutation rate was noted in the cases of NSP6 and S proteins
[57]	−	−	2443	−	45,000 genomes were downloaded from GISAID database. However, 2443 sequences with patient status were available and only 102 were analyses (56 in severe group (defined as “severe”, “ICU”, “died”) and 46 in mild group (defined as “mild”, “asymptomatic” and “not hospitalized”))	103 mutations in the mildly affected group (37 silent and 66 missense mutations) and 111 mutations (40 silent and 71 missense mutations) in the severely affected group were identified, including A1812D, L3606F, P4715L, D614G, A879S, Q57H, L84S, S194L, S202N, R203K and G204R variants	Spike protein D614G and RdRp P323L mutations in SARS-CoV-2 were associated with severity of COVID-19
[58]	−	December 2019 to 26 June 2020	3205 whole-genome sequences	−	3205 whole-genome sequences were collected from GISAID database	Phylogenetic analysis revealed four well-resolved clades (G, GH, GR and L.S.O.V)	Clade GR associated with a high mortality rate. Clades G and GH have intermediate mortality rates
[59]	−	−	90,000 genome sequences	−	692 SARS-CoV-2 genomic sequences originating from the USA, India, Italy, France and Spain downloaded from GISAID database	D614G variant	D614G variant was positively associated with case severity
[60]	−	−	69,571		69,571 SARS-CoV-2 sequences isolated in 100 different countries from six geographic areas obtained from GISAID database	Lineage G (G, GH and GR), characterized by the D614G mutation of Spike protein	The SARS-CoV-2 variant lineage G (S-D614G) was associated with increased disease severity of COVID-19
[61]	UK	1 October 2020 to 29 January 2021	54,906 paired patients from community-based COVID-19 testing centers	−	SARS-CoV-2 positive test results as S gene positive (compatible with previous variants) when cycle threshold values were: S gene <30, N gene <30, and ORF1ab gene <30. B.1.1.7 variant was classified as S gene negative, when cycle threshold values were: S gene not detected, N gene <30, and ORF1ab gene <30	B.1.1.7 variant	The B.1.1.7 variant was associated with 1.64-fold increase in mortality hazard ratio
[62]	UK	16 November 2020 to 11 January 2021	184,786 patients with clinical data retired from OpenSAFELY electronic health records	−	−	B.1.1.7 and non-B.1.1.7 variants	Relative hazard of death was higher in the patients infected with B.1.1.7 variant. This variant was significantly associated with higher absolute risk of death 28 days after a SARS-CoV-2 positive test
[63]	UK	1 November 2020 to 14 February 2021	1,146,534 patients retired from community dataset	−	Bigdata analysis from datasets provided by Public Health England	B.1.1.7 and non-B.1.1.7 variants	Hazard of death associated with B.1.1.7 is 61% (42–82%) higher than with pre-existing variants
[64]	−	23 December 2019 to 21 March 2020	1962 SARS-CoV-2 genomes	−	SARS-CoV-2 genomes were retired from 2019nCoVR. The COVID-19 infection, mortality, and recovery rates were collected from the USA Centers for Disease Control and Prevention (CDC) and virusncov	ORF1ab (p.5828P > L and p.5865Y > C), NSP13 (P504L and Y541C)	ORF1ab variation (p.5828P > L and p.5865Y > C) and NSP13 variation (P504L and Y541C) were associated with high infection and fatality rates
[65]	−	−	27,304	−	27,304 SARS-CoV-2 sequences were downloaded from GISAID database	Clade G A23403G (S:D614G) variant	Clade G viruses was associated with higher death rates
[66]	UK	1 November 2020 to 27 January 2021	198,420 patients extracted from a large primary care (QResearch), the national critical care (ICNARC CMP) and the COVID-19 testing (PHE) database		B.1.1.7 variant was classified as S-gene molecular diagnostic assay failure	B.1.1.7 variant	Patients with B.1.1.7 variant were at increased risk of critical care admission and mortality compared with patients without. For patients receiving critical care, mortality appears independent of virus strain
[67]	−	Up to 13 November 2020	2213		2213 complete genomes were downloaded from NCBI and GISAID databases. Clinical data were available for 118 patients	Seven frequent mutations resulting in dN substitutions were identified	P323L, D614G, R203K and G204R substitutions were associated with disease severity
[68]	USA	1 January to 5 April 2021	9765 SARS-CoV-2 specimens		SARS-CoV-2 specimens were sequenced at the Public Health Laboratory or the Pandemic Response Laboratory	B.1.526 (3679 patients), B.1.1.7 (1815 patients) and non-variant of concern (VOCs)/variant of interest (VOIs) variants	No difference in hospitalization or death rate was observed between B.1.526 and non-VOI/VOC variants. Patients infected with the B.1.1.7 variant were more likely to be hospitalized than those with non-VOI/VOC infections
[69]	USA	4 January to 20 March 2021	327		Data analyzed 327 COVID-19 B.1.427/B.1.429 cases retired from Colorado Department of Public Health and Environment database	B.1.427/B.1.429	B.1.427/B.1.429 more frequently cause discernible and severe illness than nationally circulating lineages do overall
2.3.3. Clinical studies							
[70]	France	March 2020 to January 2021	740	Nasopharyngeal swabs	Next-generation sequencing using Illumina technology with the Illumina Nextera XT Paired end strategy on a MiSeq instrument (for Clade 20A, Marseille-1 and Marseille-4 variants identification) and real-time PCR for N501Y variant identification	Clade 20A (254 patients), Marseille-1 (85 patients), Marseille-4 (190 patients) and N501Y (211 patients) variants	A lower rate of hospitalization associated with N501Y variant infection as compared to Clade 20A and Marseille-4 variant. A higher hospitalization rate was associated with Marseille-4 variant and Clade 20A infection as compared to Marseille-1 variant
[71]	USA	11 March to 22 April 2020	302	Nasopharyngeal swabs	Genomic sequences were constructed for each isolate according to variants called from sequence reads and the reference sequence (NC_045512.2). Multiple sequence alignments were performed using MAFFT software version 7.0. SARS-CoV-2 clade assignment followed GISAID clade guidelines and lineage nomenclature	6 different viral clades circulated; G, GR, and GH (clade group 2) represented 84.4% (255 of 302) of all identified isolates. The remainder included V, S, and Wuhan clades (clade group 1)	Clade group 1 infection was associated with higher mortality than clade group 2
[72]	UAE	29 January to 30 June 2020	256	Nasopharyngeal swabs	RNA libraries were prepared using the TruSeq Stranded Total RNA Library kit from Illumina. Libraries were sequenced using the NovaSeq SP Reagent kit from Illumina	115 patients had SARS-CoV-2 sequences. A total of 986 mutations were identified in 115 genomes, 272 were unique (majority were missense, n = 134) and 20/272 mutations were novel	A missense (Q271R) and synonymous (R41R) mutation in the S and N proteins, respectively, were identified in 2/27 patients with severe COVID-19 but not in patients with mild or moderate disease (0/86); *p* = 0.05
[73]	Brazil	January to February 2021	68	Oro/nasopharyngeal swab	Sequencing libraries were prepared using the CleanPlex SARS-CoV-2 panel (Paragon Genomics, Hayward, CA, USA) protocol. The resulting libraries were pooled in equimolar amounts and sequenced in Illumina MiSeq (Illumina, San Diego, CA, USA)	P.1 variant	P.1 variant was associated with increase in COVID-19 cases and hospitalization rate

## Data Availability

The datasets generated during and/or analyzed during the current study are available from the corresponding author, P.G., upon reasonable request.

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
