# Peer review of "SARS-CoV-2 Infectivity and Severity of COVID-19 According to SARS-CoV-2 Variants: Current Evidence"

_jcm, 2021, doi:10.3390/jcm10122635_

Round 1
Reviewer 1 Report
The premise of the paper is timely and a useful contribution to the field. The search strategy and methods for screening papers are robust and well documented.
However, I would suggest to the authors to amend their search to articles published later than their cutoff of Feb 2021, as relatively little was known about recent variants of concern at that time. A great number of studies on SARS-CoV-2 variants of concern (VOCs) have been reported in the past 3 months, and it would greatly enhance the appeal of the paper to include them.
As for the results and discussion, it is my opinion that the reporting of findings is superficial. Whereas the authors write only, for example, "Fourteen studies showed that certain viral mutations correlated with a higher infectivity than the wild-type . Only three studies showed no correlation between viral load and diverse mutational events" they should discuss the technical aspects of the studies that explain the different conclusions, what were the specific viral mutations, and provide their conclusion when all data are considered together.
Author Response
The premise of the paper is timely and a useful contribution to the field. The search strategy and methods for screening papers are robust and well documented.
Answer: Thank you very much for your feedback.
However, I would suggest to the authors to amend their search to articles published later than their cutoff of Feb 2021, as relatively little was known about recent variants of concern at that time. A great number of studies on SARS-CoV-2 variants of concern (VOCs) have been reported in the past 3 months, and it would greatly enhance the appeal of the paper to include them.
Answer: We conducted now the novel research with the cutoff at May 31, 2021. A total of 63 articles were added.
As for the results and discussion, it is my opinion that the reporting of findings is superficial. Whereas the authors write only, for example, "Fourteen studies showed that certain viral mutations correlated with a higher infectivity than the wild-type. Only three studies showed no correlation between viral load and diverse mutational events" they should discuss the technical aspects of the studies that explain the different conclusions, what were the specific viral mutations, and provide their conclusion when all data are considered together.
Answer: Thank you for your suggestions. We detailed the technical aspect of the included studies in the results section and developed now our discussion.
Reviewer 2 Report
Loi Dao et al resent a review on studies of SARS-CoV-2 infectivity and disease severity. The aim of the study is good and relevant. However, there are some concerns:
- Table 1 is too long and too complicated to read and understand. The data presented in Table 1 need to be structured in a more comprehensive way, preferably in a number of tables. I would recommend the authors to present studies on infectiveity first in a number of tables and then studies on severity in a number of tables. I would recommend that different types of studies are presented in groups separately, i.e. in vitro/animal model studies, clinical studies, and studies of sequences downloaded from GISAID database.
- Was Figure 2 developed by the authors?
- It would be useful to have a more extensive discussion about the infectivity and severity of the new variants of concern, i.e. B.1.1.7, B.1.351, P.1, B.1.427, and B.1.429.
- It would be useful if the authors could clarify where mutations and deletions are presented in the new variants of concern, i.e. B.1.1.7, B.1.351, P.1, B.1.427, and B.1.429.
Author Response
Loi Dao et al resent a review on studies of SARS-CoV-2 infectivity and disease severity. The aim of the study is good and relevant.
Answer: Thank you very much for your feedback.
However, there are some concerns:
Table 1 is too long and too complicated to read and understand. The data presented in Table 1 need to be structured in a more comprehensive way, preferably in a number of tables. I would recommend the authors to present studies on infectiveity first in a number of tables and then studies on severity in a number of tables. I would recommend that different types of studies are presented in groups separately, i.e. in vitro/animal model studies, clinical studies, and studies of sequences downloaded from GISAID database.
Answer: Thank you for your suggestions. We divided now the results into two tables, one for relations between SARS-CoV-2 variants with infectivity, and another for their relations with severity of COVID-19 diseases. We have classified the included articles into different subheadings.
Was Figure 2 developed by the authors?
Answer: Yes, we developed this Figure and it has been updated in this revised version
It would be useful to have a more extensive discussion about the infectivity and severity of the new variants of concern, i.e. B.1.1.7, B.1.351, P.1, B.1.427, and B.1.429.
Answer: We developed now our discussions.
It would be useful if the authors could clarify where mutations and deletions are presented in the new variants of concern, i.e. B.1.1.7, B.1.351, P.1, B.1.427, and B.1.429.
Answer: The mutations and deletions in the VOCs were detailed now in the table 1.